# COVID-19 vaccines and mental distress

**Francisco Perez-Arce**[1]*, **Marco Angrisani**[2], **Daniel Bennett**[2], **Jill Darling**[2], **Arie Kapteyn**[2], **Kyla Thomas**[2]

1 Center for Economic and Social Research, University of Southern California, Washington, District of Columbia, United States of America, 2 Center for Economic and Social Research, University of Southern California, Los Angeles, California, United States of America

* perezarc@usc.edu

## Abstract

### Background

The COVID-19 pandemic brought about large increases in mental distress. The uptake of COVID-19 vaccines is expected to significantly reduce health risks, improve economic and social outcomes, with potential benefits to mental health.

### Purpose

To examine short-term changes in mental distress following the receipt of the first dose of the COVID-19 vaccine.

### Methods

Participants included 8,003 adults from the address-based sampled, nationally representative Understanding America Study (UAS), surveyed at regular intervals between March 10, 2020, and March 31, 2021 who completed at least two waves of the survey. Respondents answered questions about COVID-19 vaccine status and self-reported mental distress as measured with the four-item Patient Health Questionnaire (PHQ-4). Fixed-effects regression models were used to identify the change in PHQ-4 scores and categorical indicators of mental distress resulting from the application of the first dose of the COVID-19 vaccine.

### Results

People who were vaccinated between December 2020 and March 2021 reported decreased mental distress levels in the surveys conducted after receiving the first dose. The fixed-effects estimates show an average effect of receiving the vaccine equivalent to 4% of the standard deviation of PHQ-4 scores (p-value<0.01), a reduction in 1 percentage point (4% reduction from the baseline level) in the probability of being at least mildly depressed, and of 0.7 percentage points (15% reduction from the baseline level) in the probability of being severely depressed (p-value = 0.06).

**Data Availability Statement:** All datasets are available from the Understanding America Study database uasdata.usc.edu.

**Funding:** A.K. the Bill & Melinda Gates Foundation INV-016365 A.K 5U01AG054580 National Institute

on Aging under grant. The funders had no role in study design, data collection and analysis, decision to publish, or preparation of the manuscript.

**Competing interests:** The authors have declared that no competing interests exist.

## Conclusions

Getting the first dose of COVID-19 resulted in significant improvements in mental health, beyond improvements already achieved since mental distress peaked in the spring of 2020.

## Introduction

The COVID-19 pandemic has affected several aspects of people's lives, including their employment and finances, health risks and opportunities to socialize, all of which can affect mental health. COVID-19 patients suffered psychological consequences [1] but mental distress issues arose in the general population as well. Several studies document elevated levels of psychological distress, including anxiety and suicidal thoughts, in many countries around the world [2]. In the US, mental health distress rose sharply early in the pandemic and then recovered partially. Mental health distress rose to a peak in April, but improved since then and by August had returned to a level comparable to that of early March [3].

Several factors contributed to the rise in mental health problems in the pandemic. Some studies have suggested that economic concerns were the most strongly associated with worsening mental health, while concerns about their own health and social distance were also correlated though less strongly [4]. The improvement in economic conditions and the release of public economic support in the form of unemployment insurance and stimulus checks may have been a factor in the recovery of mental health since April 2020. Studies show that sleep problems were common during the COVID-19 crisis and this was associated with depression among the general population [5].

A growing literature studies the determinants of vaccine hesitancy and willingness to get vaccinated [6–8]. Factors in the willingness to get vaccinated include age [9], sources of information [10], fear of COVID-19 [11] and perceived severity of COVID-19 [12]. In earlier studies, fear of COVID-19 was associated with increased increased future career anxiety and decreased job satisfaction [13]. A study by Kejriwal and Shen [14] found a positive correlation between willingness to get vaccinated hesitancy and negative affect (in particular, those reporting more worry and anxiety reported more willingness to vaccinate). To the extent that those who were more anxious about COVID-19 get vaccinated, the vaccine rollout may have improved mental health by reducing that anxiety. Vaccine uptake may improve quality of life and economic outlook, enabling people to resume previous activities, become more socially active, return to working in person, or become employed.

In this paper, we focus on the direct and short-term effects of being vaccinated on mental health in the first few months of the rollout, by estimating fixed-effects models that allow us to compare change over time in the mental health of those who received a vaccine compared to those who did not receive a vaccine. We note that there could be indirect effects too, which we do not study here. Indirect effects would arise, for instance, through the reduction in risk for those who are not vaccinated but stand to benefit from increased herd immunity.

Studying how mental health evolves as the country recovers from the pandemic can shed light on the relationship between mental health and pandemic-related stressors.

## Methods

The Understanding America Study (UAS) is a nationally representative longitudinal study of adult Americans 18 and over. Respondents are recruited through address-based sampling from the U.S. Postal Service Delivery Sequence files. Respondents without internet access are

provided with a tablet, internet access and training on how to use the tablets if necessary. UAS respondents are paid $20 per 30 minutes of survey time [15].

UAS panelists were invited to participate in a bi-weekly tracking survey to understand the impacts of the pandemic, which we named the Understanding Coronavirus in America Study (UCAS) [16]. The University of Southern California Institutional Review Board reviewed and approved the study (UP-14-00148-AM088). Respondents provided written informed consent. All participants were 18 years of age or older. On March 10, 2020, panelists were invited to answer the first survey (which remained open until the end of March). Between April 1 2020 and February 16 2021, UCAS participants were invited to answer surveys every fourteen days. This frequency was chosen to allow tracking how people's perceptions, behaviors and outcomes evolved throughout the pandemic. After February 16, the bi-weekly cycle was replaced by a four-week cycle, so that since then respondents answer questions every four weeks. Participants were randomly assigned a number between one and fourteen, which determined the day on which they were asked to answer the survey. Upon invitation, the respondent had two weeks to complete the survey. Variables measured every wave include PHQ-4 scores and COVID-19 vaccination status (the latter since December 23, 2020).

Data from every UCAS wave are made available to the research community on the day after the field period closes (https://uasdata.usc.edu/covid19). Questionnaires are available at *https://uasdata.usc.edu/page/Covid-19+Documentation*. The 25[th] wave of the survey closed on March 30, 2021. We use all data from surveys completed by March 31[st] of 2021, which includes partial data from the 26[th] wave. Overall, our dataset spans the period from March 10, 2020 to March 31[st], 2021 [17].

From the 8,955 UAS panelists who were invited to participate in UCAS, 97.1% agreed to participate in UCAS, and 94% answered at least one survey. Across all waves, the response rate was 82% on average. The sample we use consists of answers from 8,027 adults who completed at least two waves of the survey. Altogether, our data comprise 157,227 respondent-wave observations.

## Measures

### Mental distress

We measure mental distress with the four-item Patient Health Questionnaire (PHQ-4) [18]. Two items measure depressive symptoms and two items measure anxiety symptoms. Responses to each item are scored from 0 to 4 and summed to create an index ranging between 0 and 16 with higher numbers indicating higher levels of mental distress.

We use the PHQ-4 score as an outcome variable, as well as three indicator variables based on the thresholds used in [18]: *mild mental distress or higher*, which takes the value of one if PHQ-4 is equal to or higher than three; *moderate mental distress or higher*, which equals one if PHQ4 if equal to or higher than six; and *severe mental distress* which equals one if PHQ-4 is equal to or higher than 9. The validity and reliability of the PHQ-4 is supported by earlier studies [19].

### Vaccination

Starting on December 23, 2020, the UCAS surveys asked panelists whether they had received their first shot of a COVID-19 vaccine. From that question, we constructed the indicator *ever vaccinated*, which equals "0," or "never vaccinated," for respondents who were never vaccinated during the study period and "1," or "ever vaccinated," for respondents who received a first dose at any point during the study period. We use this indicator first in our analysis to

graph the mental health trajectory of respondents who were vaccinated at some point during the study period and compare it to the trajectory of respondents who were never vaccinated.

For our fixed-effects regression analyses, we constructed the point-in-time indicator *has vaccine*, which, at any given point in time, equals "0," or "not vaccinated," for any individual who has not received the first dose of the vaccine and "1," or "vaccinated," once an individual indicates that they have received the first dose. For respondents who were not vaccinated by March 2021, *has vaccine* equals "0" at all time points.

### Auxiliary variables

We use demographic indicators for race and ethnicity, educational attainment and gender to study heterogeneity of effects.

## Statistical analysis

### Trajectories of mental distress over time: Ever vs. never vaccinated

We begin by comparing mental distress trajectory of respondents who received the vaccine at some point during the study period (*ever vaccinated* = 1) with mental distress trajectory of respondents who never received the vaccine during the study period (*ever vaccinated* = 0). For each group, we estimate mental distress trajectories using average PHQ-4 scores. We estimate these trajectories using a local polynomial approximation of the date, measured as days passed since the first date in the panel (March 10, 2020) using the *lpoly* function in STATA.

### Fixed-effects regression analysis of effect of vaccination on mental distress

We estimated regression models as in Eq 1 below, where $Y_{it}$ is the outcome variable of interest (*mental distress*) for individual $i$ in survey wave $t$, $\alpha_i$ is an individual fixed effect (to capture differences across subjects, which may correlate with vaccination status), $\tau_t$ are survey wave fixed effects (to capture differences across time which are common among those vaccinated and not), and $Vacc_{it}$ is the indicator of whether the individual has been vaccinated by survey wave t (*has vaccine*). $\beta$ is the coefficient of interest, and represents the association of *has vaccine* with *mental distress* after accounting for idiosyncratic differences of those who got vaccinated as well as for common time effects.

$$Y_{it} = \alpha_i + \tau_t + \beta Vacc_{it} + \varepsilon_{it} \tag{1}$$

In further specifications, $Y_{it}$ is an indicator for the three thresholds for mental distress: *mild mental distress or higher*, *moderate mental distress or higher*, and *severe mental distress*. In all cases, we cluster standard errors at the individual level.

To analyze heterogeneity in the impact of vaccination on different groups, we estimated equations like (1) above but separately for different groups (men and women, college educated and not-college educated, White and non-White). Let $\beta^A$ denote the coefficient for *has vaccine* for group A (for instance, women) and $\beta^B$ for group B (for instance, men). For pairs of groups A and B, we performed Wald tests to determine whether the hypotheses $\beta^A = \beta^B$ can be rejected.

### Inference

Statistical significance was assessed at the $p < .05$ level. Analyses were conducted using Stata version 14 (StataCorp Inc., College Station, TX). Data was analyzed in April of 2021.

## Results

### Descriptive statistics

As shown in Table 1, the *ever vaccinated* and *never vaccinated* groups differ significantly in their demographic composition as well as in baseline levels of mental health. The differences are likely a result both from the vaccine eligibility rules applicable during the period that we study as well as different levels of vaccine enthusiasm or hesitancy across demographic groups. Notable differences include: mean age (60.4 years among the vaccinated, 47.1 among the not vaccinated, p-value of the difference<0.01), education level (68% college educated in the vaccinated and 52% in the not vaccinated groups, p-value<0.01), race and ethnicity (87% White among the vaccinated, 6% Black, 11% Hispanic among the vaccinated, 82% White, 11% Black and 18% Hispanic among the non-vaccinated, in all cases p-value of differences <0.01). While we do not have data on the occupation of respondents, it is likely that the *ever vaccinated* group contains a larger percentage of health care and other essential workers who were prioritized in the vaccination rollout.

### Trajectory of mental distress over time: Ever vs. never vaccinated

Fig 1 shows the trajectory of average PHQ-4 scores among the *ever vaccinated* and the *never vaccinated* groups. While the levels differ at baseline, the trajectories are similar across the two groups until around December 2020 when the vaccines became available. From March 10 2020 ("day 0") onward mental distress increased sharply during the first 30 days, and then recovered. For both groups, it reached the level of March 10 2020 before day 100, and remained fairly stable until vaccine rollout started for both groups. After that, we see a divergence of trajectories until the last day of the study period.ppp

We observe that the never-vaccinated group exhibits a higher PHQ-4 score than the ever-vaccinated group throughout the study period. Although a detailed explanation of this difference is beyond the scope of this paper, we note that at least part of the difference can be ascribed to the different composition of the groups. For instance, on average, individuals over

**Table 1. Backround characteristics.**

|  | (1) Never Vaccinated | (2) Ever Vaccinated | (1) vs. (2), p-value |
|---|---|---|---|
| Average Age | 47.12 | 60.37 | <0.001 |
|  | (0.21) | (0.42) |  |
| Percentage Male | 40.4% | 44.3% | 0.008 |
|  | (0.7) | (1.30 |  |
| Percentage with college degree | 51.9% | 68.4% | <0.001 |
|  | (0.7) | (1.2) |  |
| Perecentage White | 81.6% | 87.2% | <0.001 |
|  | (0.5) | (0.9) |  |
| Percentage Black | 10.9% | 5.9% | <0.001 |
|  | (0.4) | (0.6) |  |
| Percentage Spanish/Hispanic/Latino | 17.9% | 11.1% | <0.001 |
|  | (0.5) | (0.8) |  |
| N | 6,384 | 1,643 |  |

Note: Unweighted sample composition by vaccination status over the study period (March 10 2020 to March 31, 2021). Vaccinated: *ever vaccinated* respondents are those who reported having received at least a dose between March 10, 2020 and March 14, 2021, *Never vaccinated* respondents are those who did not report receiving a vaccine dose between March 10, 2020 and March 14, 2021.

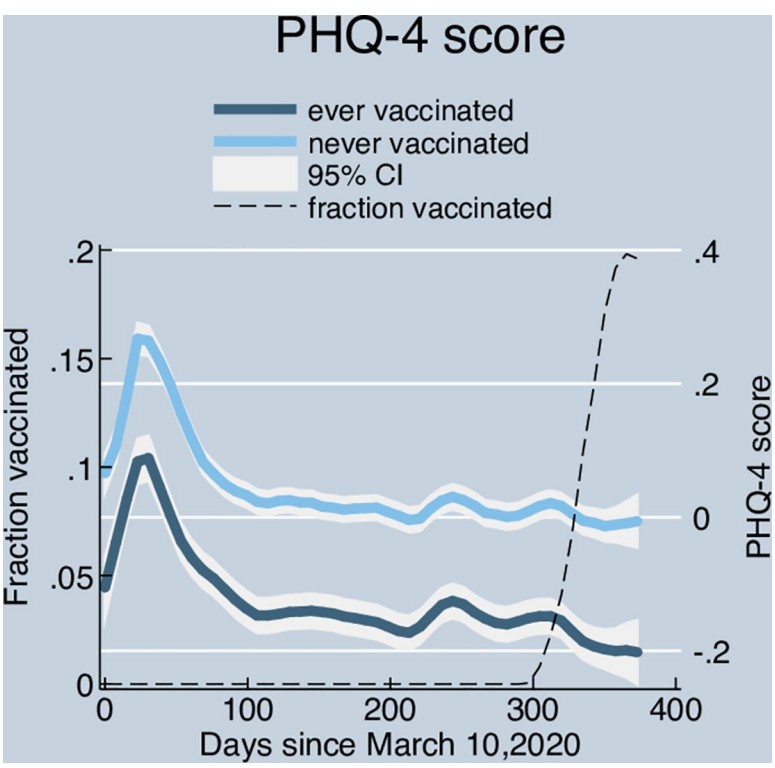

**Fig 1. Trajectory of mental distress over time by vaccination group.** Note. PHQ-4 scores are standardized to have a mean of 0 and a standard deviation of 1. *Ever vaccinated* respondents are those who reported having received at least a dose by March 14, 2021, *Never vaccinated* respondents are those who did not report receiving a vaccine dose between by March 14, 2021. Local polynomial approximation to date (days since the initial panel date-March 10, 2020).

65 have systematically shown better mental health during the pandemic than younger age groups (https://covid19pulse.usc.edu).

For the *ever-vaccinated*, Fig 2 below shows the change in standardized PHQ-4 scores by number of days before and after receiving the first dose. In order to abstract from common time effects, it uses residuals from a regression of PHQ-4 scores on wave dummy variables. The residuals are plotted against the number of days before and after receiving the first dose. The graph shows PHQ-4 scores fall after receiving the first vaccination dose.ppp

## Fixed effects regression analysis of the effect of vaccination on mental distress

As Table 2 shows, the coefficient β, our estimate of the effect of receiving at least one dose of the vaccine (*has vaccine* = 1) on standardized PHQ-4 scores, equaled -0.04 (p-value<0.01), so that receiving a vaccine dose reduced PHQ-4 scores by 4% of a standard deviation. Receiving the vaccine was associated with a 1 percentage point decrease (4% from the baseline level) in the probability of being at least mildly depressed (p-value = 0.06); and a 0.7 percentage point (15% from the baseline level) decrease in the probability of being severely depressed (p-value = 0.01). The effect on the probability of experiencing moderate mental distress was non-significant (p-value = 0.26).

In the supplementary materials, we provide estimates from a model where we include an interaction of $Vacc_{it}$ with time passed since first reporting a vaccine dose. The results show that the effect does not fade out as time passes (at least within the study period). On average,

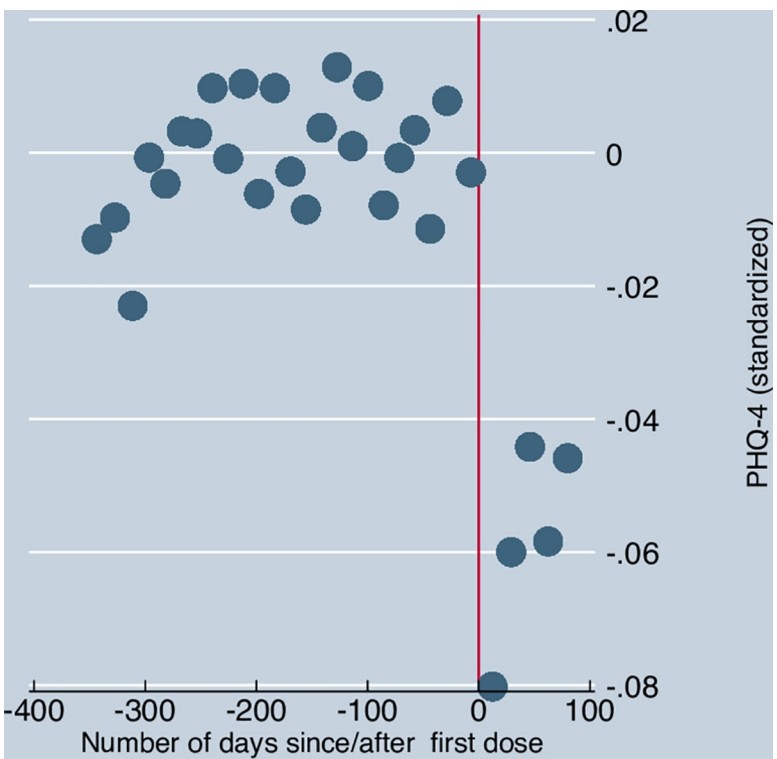

**Fig 2. Mental distress before and after receiving the first dose.** Note. PHQ-4 scores are standardized to have a mean of 0 and a standard deviation of 1. Residuals are taken from a linear regression of standardized PHQ-4 scores against wave dummy variables and an indicator for *ever vaccinated*. For the *ever vaccinated*, the residual is plotted against the number of days before or since receiving the first dose. Observations are grouped by wave. Vertical line denotes the date when a vaccination dose was first reported.

the standardized PHQ-4 score fell by 0.03 standard deviations(p-value<0.01) immediately after vaccination and was further reduced by 0.003 standard deviations per ten days after vaccination, although not significantly (p-value = 0.27)

**Table 2. Fixed effects regression models.**

|  | PHQ-4 Score (standardized) | Mild mental distress or higher[1] | Moderate mental distress or higher[2] | Severe mental distress[3] |
|---|---|---|---|---|
| Has vaccine | -0.0352*** | -0.0104* | -0.00422 | -0.00681** |
| Standard error | (0.0109) | (0.00548) | (0.00372) | (0.00272) |
| P-value | 0.001 | 0.059 | 0.256 | 0.012 |
| Mean dependent variable | 0.000 | 0.291 | 0.106 | 0.0446 |
| Observations | 157,228 | 157,228 | 157,228 | 157,228 |
| R-squared | 0.722 | 0.617 | 0.530 | 0.506 |

Respondent fixed effects and survey-wave dummies included in the regression. Standard errors clustered at the individual level

*** p-value<0.01

** p-value<0.05

* p-value<0.

[1] *Mild mental distress or higher* is an indicator variable that takes the value of one if PHQ-4 is equal to or higher than three and 0 otherwise

[2] *Moderate mental distress or higher* is an indicator variable that takes the value of one if PHQ-4 is equal to or higher than six and 0 otherwise

[3] *Severe mental distress* is an indicator variable that takes the value of one if PHQ-4 is equal to or higher than six and 0 otherwise.

## Heterogeneity

Table 3 shows the results of the heterogeneity analyses. The effect on standardized PHQ4-scores is found to be higher for women, $\beta^{Female}$ = -0.4, p-value <0.01, than for men, $\beta^{Male}$ = -0.02, p-value = 0.08, although we cannot rule out that the effects are equal for men and women (p-value for the test of $\beta^{Female} = \beta^{Male}$ equals 0.36).

The effects for *mild mental distress or higher, and severe mental distress* are statistically significant for women but not for men. The coefficients imply that getting vaccinated reduced the probability of being at least mildly depressed for women by 1.6 percentage points (5% reduction compared to the base value, p-value<0.01 and the probability of being severely depressed or higher by 0.8 percentage points (15% reduction, p-value<0.01)

The coefficients for PHQ-4 scores for both White and non-White respondents were negative and insignificantly different across the two groups ($\beta^{White}$ = -0.03, $\beta^{Non-white}$ = -0.04,

**Table 3. Heterogeneity analysis.**

Panel A. Gender

|  | PHQ-4 Score (standardized) | Mild mental distress or higher[1] | Moderate mental distress or higher[2] | Severe mental distress[3] |
|---|---|---|---|---|
| Has vaccine X Male | -0.025* | -0.003 | -0.006 | -0.005 |
| Standard error | (0.014) | (0.008) | (0.005) | (0.004) |
| Has vaccine X Female | -0.044*** | -0.016** | -0.003 | -0.008** |
| Standard error | (0.016) | (0.008) | (0.005) | (0.004) |
| p-val ($\beta^{Female} = \beta^{Male}$) | 0.357 | 0.220 | 0.721 | 0.551 |
| Observations | 157,227 | 157,227 | 157,227 | 157,227 |
| R-squared | 0.722 | 0.617 | 0.530 | 0.506 |

Panel B. Race

|  | PHQ-4 Score (standardized) | Mild mental distress or higher | Moderate mental distress or higher | Severe mental distress |
|---|---|---|---|---|
| Has vaccine X White | -0.033*** | -0.013** | -0.002 | -0.005 |
| Standard error | (0.012) | (0.006) | (0.004) | (0.003) |
| Has vaccine X Non-white | -0.044* | -0.0032 | -0.012 | -0.013** |
| Standard error | (0.0233) | (0.0112) | (0.007) | (0.005) |
| p-val ($\beta$White = $\beta$Non-white) | 0.680 | 0.446 | 0.224 | 0.174 |
| Observations | 156,987 | 156,987 | 156,987 | 156,987 |
| R-squared | 0.722 | 0.617 | 0.529 | 0.506 |

Panel C. Education

| Gender | PHQ-4 Score | Mild mental distress or higher | Moderate mental distress or higher | Severe mental distress |
|---|---|---|---|---|
| Has vaccine X College Educated | -0.032** | -0.016** | -0.002 | -0.004 |
| Standard error | (0.013) | (0.007) | (0.005) | (0.003) |
| Has vaccine X Not college educated | -0.038* | -0.001 | -0.006 | -0.012** |
| Standard error | (0.019) | (0.009) | (0.00657) | (0.00526) |
| p-val ($\beta^{College} = \beta^{Non-college}$) | 0.797 | 0.203 | 0.709 | 0.221 |
| Observations | 157,186 | 157,186 | 157,186 | 157,186 |
| R-squared | 0.722 | 0.617 | 0.529 | 0.506 |

Respondent fixed effects and survey-wave by group dummies included in the regression. Standard errors clustered at the individual level.

*** p-value<0.01

** p-value<0.05

* p-value<0.1.

[1] *Mild mental distress or higher* is an indicator variable that takes the value of one if PHQ-4 is equal to or higher than three and 0 otherwise

[2] *Moderate mental distress or higher* is an indicator variable that takes the value of one if PHQ-4 is equal to or higher than six and 0 otherwise

[3] *Severe mental distress* is an indicator variable that takes the value of one if PHQ-4 is equal to or higher than six and 0 otherwise.

p-value of difference = 0.68). They were also not significantly different for college educated versus non-college educated respondents ($\beta^{College}$ = -0.03 $\beta^{Non-college}$ = -0.04, p-value of the difference = 0.80). These results suggest that the improvements in mental health following vaccination were not circumscribed within specific racial or education attainment groups.

## Discussion

Earlier work showed that the prevalence of mental distress peaked in mid-April to early May 2020 and declined thereafter; it also showed how those trajectories differed for demographic groups. In this paper, we document how mental health distress has diverged between those who have been vaccinated and those who have not (or at least not yet). By comparing the trajectories of these two groups, we learn about the short-term impact of vaccination on mental health.

The results here should be interpreted as the short-term direct effects of getting a first vaccine dose. The overall contribution of vaccine uptake on improving mental health outcomes is potentially much larger, as it affects not only those vaccinated but also the unvaccinated. An unvaccinated individual may still benefit from the reduced prevalence rates in the population, may become less worried about loved ones, and may benefit from increased social and economic opportunities if the vaccine rollout results in more social and economic activity due to lower disease risk.

There are some limitations to this research. In particular, it is possible that the difference in trajectories across the vaccinated and not vaccinated groups arose not due to a causal effect of receiving the vaccine dose but from sorting at the time of the vaccine rollout, such that individuals with an increased likelihood of becoming less depressed were also more likely to decide to get vaccinated. In order to investigate that possibility, one can take advantage of different dates at which individuals in different groups have become eligible for vaccination. We leave that to future work.

The effects we identify could arise from one of or a combination of mechanisms. Those recently vaccinated may become less worried about getting infected, they may become more active socially, or they may venture into different work opportunities. Future research should investigate the mechanisms through which the vaccine shot achieved such effects.

The vaccination effects are likely to be heterogeneous on characteristics beyond the ones analyzed here. Since people who get the vaccines at different times are different in several dimensions, this implies that the effects may be different for the people who get vaccinated after the period studied here. Another reason why the effects may be different in a later period is that the conditions may be different. For instance, if COVID-19 cases are substantially reduced, then getting the vaccine may have a lower impact on the vaccinee's health concerns. Whether that is the case can be studied at a later date using a methodology similar to the one used here.

## Supporting information

**S1 Appendix. Effects as a function of time elapsed since first vaccination.**
(PDF)

## Author Contributions

**Conceptualization:** Francisco Perez-Arce, Marco Angrisani, Daniel Bennett, Jill Darling, Arie Kapteyn.

**Data curation:** Francisco Perez-Arce.

**Formal analysis:** Francisco Perez-Arce, Kyla Thomas.

**Funding acquisition:** Arie Kapteyn.

**Methodology:** Francisco Perez-Arce, Daniel Bennett.

**Supervision:** Arie Kapteyn.

**Visualization:** Daniel Bennett.

**Writing – original draft:** Francisco Perez-Arce.

**Writing – review & editing:** Francisco Perez-Arce, Marco Angrisani, Daniel Bennett, Jill Darling, Arie Kapteyn, Kyla Thomas.

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
