## [Decision Letter · Decision Letter 0]

24 Jun 2021

PONE-D-21-16421

COVID-19 Vaccines and Mental Distress

PLOS ONE

Dear Dr. Perez-Arce,

Thank you for submitting your manuscript to PLOS ONE. After careful consideration, we feel that it has merit but does not fully meet PLOS ONE’s publication criteria as it currently stands. Therefore, we invite you to submit a revised version of the manuscript that addresses the points raised during the review process.

An expert in this field and I have reviewed your submission. We both agree that your contribution may add something to the literature. However, there are some essential corrections needed before I reevaluate your contribution. Please pay attention to all the comments made by the reviewer and answered them using a point-by-point response letter. Apart from his comments, I would like you to consider my following comments as well.

1. There is a growth of literature on vaccine hesitancy and the relationship between vaccine and mental distress. However, the present study does not provide a comprehensive literature review in this content. The authors are thus recommended to consult the following references to strengthen their literature review.

Rajabimajd N, Alimoradi Z, Griffiths MD. Impact of COVID-19-related fear and anxiety on job attributes: A systematic review. Asian J Soc Health Behav 2021;4:51-5

Patil ST, Datar MC, Shetty JV, Naphade NM. “Psychological consequences and coping strategies of patients undergoing treatment for COVID-19 at a tertiary care hospital”: A qualitative study. Asian J Soc Health Behav 2021;4:62-8

Rieger MO. Willingness to vaccinate against COVID-19 might be systematically underestimated. Asian J Soc Health Behav 2021;4:81-3

Rieger MO. Triggering altruism increases the willingness to get vaccinated against COVID-19. Soc Health Behav 2020;3:78-82

Yahaghi, R., Ahmadizade, S., Fotuhi, R., Taherkhani, E., Ranjbaran, M., Buchali, Z., Jafari, R., Zamani, N., Shahbazkhania, A., Simiari, H., Rahmani, J., Yazdi, N., Alijani, H., Poorzolfaghar, L., Rajabi, F., Lin, C.-Y., Broström, A., Griffiths, M. D., Pakpour, A. H. (2021). Fear of COVID-19 and perceived COVID-19 infectability supplement Theory of Planned Behavior to explain Iranians’ intention to get COVID-19 vaccinated. Vaccines, 9, 684.

Wang, P.-W., Ahorsu, D. K., Lin, C.-Y., Chen, I.-H., Yen, C.-F., Kuo, Y.-J., Griffiths, M. D., & Pakpour, A. H. (2021). Motivation to Have COVID-19 Vaccination Explained Using an Extended Protection Motivation Theory Among University Students in China: The Role of Information Sources. Vaccines, 9, 380. 

Yeh, Y.-C., Chen, I.-H., Ahorsu, D. K., Ko, N.-Y., Chen, K.-L., Li, P.-c., Yen, C.-F., Lin, C.-Y., Griffiths, M. D., Pakpour, A. H. (2021). Measurement invariance of the Drivers of COVID-19 Vaccination Acceptance Scale: Comparison between Taiwanese and mainland Chinese-speaking populations. Vaccines, 9(3), 297. 

Kukreti, S., Lu, M.-L., Lin, Y.-H., Strong, C., Lin, C.-Y., Ko, N.-Y., Chen, P.-L., & Ko, W.-C. (2021). Willingness of Taiwan’s healthcare workers and outpatients to vaccinate against COVID-19 during a period without community outbreaks. Vaccines, 9(3), 246. 

Alimoradi, Z., Broström, A., Tsang, H. W. H., Griffiths, M. D., Haghayegh, S., Ohayon, M. M., Lin, C.-Y., Pakpour, A. H. (2021). Sleep problems during COVID-19 pandemic and its’ association to psychological distress: A systematic review and meta-analysis. EClinicalMedicine, 36, 100916. 

Pramukti, I., Strong, C., Sitthimongkol, Y., Setiawan, A., Pandin M. G. R., Yen, C.-F., Lin, C.-Y., Griffiths, M. D., Ko, N.-Y. (2020). Anxiety and suicidal thoughts during the COVID-19 pandemic: A cross-country comparison among Indonesian, Taiwanese, and Thai university students. Journal of Medical Internet Research, 22(12), e24487. 

2. The meaning of "Fraction" on Table 1 is unclear. Moreover, what is (1) and (2) in Table 1? Please do not report p at 0.000 because p-value can only be very small but never be 0. Use p < 0.001 instead. The values in the columns of "Never Vaccinated" and "Ever Vaccinated" in Table 1 are unclear what they are.

3. In Table 2, please define what "mild mental distress or higher", "moderate mental distress or higher", and "severe mental distress" are. It is unclear what "mean dependent variable" is in Table 2. 

4. In Table 3, please also define what "mild mental distress or higher", "moderate mental distress or higher", and "severe mental distress" are.

We look forward to receiving your revised manuscript.

Kind regards,

Chung-Ying Lin

Academic Editor

PLOS ONE

Journal Requirements:

2. Please include additional information regarding the survey or questionnaire used in the study and ensure that you have provided sufficient details that others could replicate the analyses. For instance, if you developed a questionnaire as part of this study and it is not under a copyright more restrictive than CC-BY, please include a copy as Supporting Information.

3. We note you have included a table to which you do not refer in the text of your manuscript. Please ensure that you refer to Table 1 and 3 in your text; if accepted, production will need this reference to link the reader to the Table.

Reviewers' comments:

Reviewer's Responses to Questions

**Comments to the Author**

1. Is the manuscript technically sound, and do the data support the conclusions?

Reviewer #1: Yes

2. Has the statistical analysis been performed appropriately and rigorously? 

Reviewer #1: N/A

3. Have the authors made all data underlying the findings in their manuscript fully available?

Reviewer #1: Yes

4. Is the manuscript presented in an intelligible fashion and written in standard English?

Reviewer #1: Yes

5. Review Comments to the Author

Reviewer #1: Q1: In paragraph 1 of introduction part, the author mentioned the dynamic of mental distress experienced of people due to COVID-19 which is up and down. It would be better to explain more why this fluctuated scheme could happen.

the authors may include the second paragraph into the first paragraph.

Q2: It is questioned that how the authors convinced the readers that vaccine really can improve well-being and mental health? How the vaccine uptake may reduce stress and alleviate fears of passing a COVID-19 infection along to others in social settings?

There is no citation at all for the 3rd par which talk about vaccine intake

Q3:

In the introduction part, there is insufficient reasons why the authors conduct this research? The author may add some crucial information about why the readers will have benefit to know the research result.

Q4:

The authors should provide the scientific reason why the panelists complete the surveys in the 14 days interval?

Q5:

There is no information about the validity and reliability of the PHQ-4?

Q6:

The authors stated “The ever vaccinated and never vaccinated groups differ significantly in their demographic composition as well as in baseline levels of mental health. The differences are likely a result both from the vaccine eligibility rules applicable during the period that we study as well as different levels of vaccine enthusiasm or hesitancy across demographic groups” in result section. Unfortunately, there is no explanation about the occupational background of the respondents, which may explain why they get vaccinated and never vaccinated. For instance the group of health care professionals.

Q7:

1. Table 1 the authors should clearly indicate the column label (number and percentage)

2. The authors need provide the information about the total number of the respondents in the table 1

Q8:

In figure 1 how the authors determined the time/date of group get vaccinated? Since the vaccination for COVID-19 started in December 23, 2020, so why the authors mentioned the interval was between March 10, 2020 and March 14, 2021?

Q9:

It was less information in discussion part that the authors may explain deeper interpretation from the result part. For instance, the comparison from each group of different educational background, and ethnics.

Q10:

There was miss-typed and word redundant

6. PLOS authors have the option to publish the peer review history of their article (what does this mean?). If published, this will include your full peer review and any attached files.

Reviewer #1: No

---

## [Author Response · Author response to Decision Letter 0]

30 Jul 2021

We are deeply grateful for your and the referee’s review and comments. We have revised the paper accordingly and believe it has substantially improved. In particular, per your first comment, we have now better situated the study in the relevant literature. 

Below, please find our responses to the individual comments.

I. Response to the Editor’s comments

An expert in this field and I have reviewed your submission. We both agree that your contribution may add something to the literature. However, there are some essential corrections needed before I reevaluate your contribution. Please pay attention to all the comments made by the reviewer and answered them using a point-by-point response letter. Apart from his comments, I would like you to consider my following comments as well.

1. There is a growth of literature on vaccine hesitancy and the relationship between vaccine and mental distress. However, the present study does not provide a comprehensive literature review in this content. The authors are thus recommended to consult the following references to strengthen their literature review.

We agree the original submission was lacking on this respect. We have now better situated the study in the relevant literature, making the connection to the literatures on vaccine hesitancy and mental health effects of COVID-19. For context, it is particularly relevant to show that fear of COVID-19, as well as anxiety in general, are predictors of willingness to vaccinate. Page 4 now discusses this literature, which includes all but one of the references suggested by you as well as a number of other references that we deemed relevant. 

2. The meaning of "Fraction" on Table 1 is unclear. Moreover, what is (1) and (2) in Table 1? Please do not report p at 0.000 because p-value can only be very small but never be 0. Use p < 0.001 instead. The values in the columns of "Never Vaccinated" and "Ever Vaccinated" in Table 1 are unclear what they are.

We made the change to use percentages instead of fractions which makes Table 1 clearer in our opinion. We also changed the reporting of p-values to p<0.0001 when applicable, and added the missing column numbers –(1) and (2)- to make the table clearer. With these changes, we hope the contents of Table 1 are now clear.

3. In Table 2, please define what "mild mental distress or higher", "moderate mental distress or higher", and "severe mental distress" are. It is unclear what "mean dependent variable" is in Table 2. 

We now define these variables in the note to Table 2.

4. In Table 3, please also define what "mild mental distress or higher", "moderate mental distress or higher", and "severe mental distress" are.

We now define these variables in the note to Table 3.

Reviewers' comments:

Reviewer's Responses to Questions

Comments to the Author

Reviewer #1: Q1: In paragraph 1 of introduction part, the author mentioned the dynamic of mental distress experienced of people due to COVID-19 which is up and down. It would be better to explain more why this fluctuated scheme could happen.

the authors may include the second paragraph into the first paragraph.

We added a discussion about a possible explanation for the fluctuation (economic deterioration and recovery) and cited a more extensive literature (page 4).

Q2: It is questioned that how the authors convinced the readers that vaccine really can improve well-being and mental health? How the vaccine uptake may reduce stress and alleviate fears of passing a COVID-19 infection along to others in social settings?

There is no citation at all for the 3rd par which talk about vaccine intake

We have added a discussion of the literature with the corresponding citations. In particular, we cited literature that showed that people with more anxiety are more likely to be willing to be vaccinated. It is then possible that getting the vaccine relieved those fears, which may be the channel through which vaccination improved mental health.

Q3:

In the introduction part, there is insufficient reasons why the authors conduct this research? The author may add some crucial information about why the readers will have benefit to know the research result.

We have added a paragraph in the first page of the introduction. The linkage to the literature suggested by the editor also helps to solve this issue. 

Q4:

The authors should provide the scientific reason why the panelists complete the surveys in the 14 days interval?

Page 6 now includes an explanation of the rationale for the periodicity of the surveys.

Q5:

There is no information about the validity and reliability of the PHQ-4?

We now cite the validation and reliability study for the PHQ-4 measure (page 7, second paragraph)

Q6:

The authors stated “The ever vaccinated and never vaccinated groups differ significantly in their demographic composition as well as in baseline levels of mental health. The differences are likely a result both from the vaccine eligibility rules applicable during the period that we study as well as different levels of vaccine enthusiasm or hesitancy across demographic groups” in result section. Unfortunately, there is no explanation about the occupational background of the respondents, which may explain why they get vaccinated and never vaccinated. For instance the group of health care professionals.

The dataset we use does not contain occupation data, unfortunately. We agree that occupations are likely to be different across groups, and in indeed the larger proportion of health care professionals among the “ever vaccinated” may indeed at least partly explain the larger percentage of respondents with a college degree in that group. We now state that in the first paragraph of page 10.

Q7:

1. Table 1 the authors should clearly indicate the column label (number and percentage).

2. The authors need provide the information about the total number of the respondents in the table 1.

The labels in columns 1 and 2 have been added and the total number of respondents is shown in the last row.

Q8:

In figure 1 how the authors determined the time/date of group get vaccinated? Since the vaccination for COVID-19 started in December 23, 2020, so why the authors mentioned the interval was between March 10, 2020 and March 14, 2021?

We agree the statement in the note of Figure 1 was confusing. We change it so that it simply says that the Ever vaccinated are those who received a dose by March 14, and the Never vaccinated are those who had not received any dose by March 14. 

Q9:

It was less information in discussion part that the authors may explain deeper interpretation from the result part. For instance, the comparison from each group of different educational background, and ethnics.

We have added a clarification to the description of the heterogeneity analysis.

Q10:

There was miss-typed and word redundant

We revised and corrected the text for typos and redundant words.

Response to Journal Requirements:

2. Please include additional information regarding the survey or questionnaire used in the study and ensure that you have provided sufficient details that others could replicate the analyses. For instance, if you developed a questionnaire as part of this study and it is not under a copyright more restrictive than CC-BY, please include a copy as Supporting Information.

The surveys are publicly available at https://uasdata.usc.edu/page/Covid-19+Documentation. We have added the link to the study. 

3. We note you have included a table to which you do not refer in the text of your manuscript. Please ensure that you refer to Table 1 and 3 in your text; if accepted, production will need this reference to link the reader to the Table.

We have added the reference to Tables 1 and 3 in the text.

---

## [Editor Report · Decision Letter 1]

6 Aug 2021

COVID-19 Vaccines and Mental Distress

PONE-D-21-16421R1

Dear Dr. Perez-Arce,

We’re pleased to inform you that your manuscript has been judged scientifically suitable for publication and will be formally accepted for publication once it meets all outstanding technical requirements.

Kind regards,

Chung-Ying Lin

Academic Editor

PLOS ONE

Additional Editor Comments (optional):

I thank the authors carefully reviewed the comments and addressed all of them in this revision.
---

## [Editor Report · Acceptance letter]

11 Aug 2021

PONE-D-21-16421R1 

COVID-19 Vaccines and Mental Distress 

Dear Dr. Perez-Arce:

I'm pleased to inform you that your manuscript has been deemed suitable for publication in PLOS ONE. Congratulations! Your manuscript is now with our production department. 

Kind regards, 

on behalf of

Dr. Chung-Ying Lin 

Academic Editor

PLOS ONE